# Evaluation of Interaction between Bridge Infrastructure Resilience Factors against Seismic Hazard

**Ángel Francisco Galaviz Román** [1], **Md Saiful Arif Khan** [1], **Golam Kabir** [1,*], **Muntasir Billah** [2] **and Subhrajit Dutta** [3]

1 Industrial Systems Engineering, University of Regina, Regina, SK S4S 0A2, Canada
2 Department of Civil Engineering, University of Calgary, Calgary, AB T2N 1N4, Canada
3 Civil Engineering Department, National Institute of Technology (NIT) Silchar, Assam 781017, India
* Correspondence: golam.kabir@uregina.ca

**Abstract:** Infrastructure systems, such as bridges, are perpetually vulnerable to natural hazards such as seismic events, flooding, and landslides. This study aims to determine the relevant parameters required to increase the seismic resilience of bridge infrastructure based on the decisions of experts and prior research. To this end, the crisp DEMATEL (decision-making and trial evaluation laboratory) and rough DEMATEL methods are employed. Rough DEMATEL is a supplement to crisp DEMATEL that incorporates rough theory to handle ambiguity. The efficacies of the rough and crisp DEMATEL methods are then compared between the two approaches. This study found the most crucial seismic-resilience variables for bridges. The outcomes of this study reveal the significant order and cause-and-effect relationships. This research can assist transportation engineers and executive agencies in enhancing the seismic resilience of roadway bridges and bridge networks.

**Keywords:** roadway bridges; seismic resiliency; seismic events; rough DEMATEL; crisp DEMATEL

## 1. Introduction

Bridges comprise a crucial element of a nation's infrastructure. As a result of global climate change, this infrastructure is unfortunately vulnerable to various hazards. For instance, seismic activity, flooding, tidal waves, and landslides have structural and operational impacts on transportation systems, limiting territories' interconnectivity, ease of access to essential services, economic output, and supplies. These effects incur costs for highway users, road authorities, and economic systems [1]. In addition to longer travel time and increased fuel consumption, users suffer non-travel–related operational expenses [2].

Seismic hazard, in particular, is a risk to bridge infrastructure. A seismic hazard creates a threat that must be controlled to mitigate the repercussions. A bridge's capacity to survive such dangers relies on the interplay of the complex systems from which it was constructed. Since natural catastrophes are unavoidable, bridge infrastructure needs to be resilient.

However, evaluating resilience is a complex endeavour involving qualitative and quantitative data from diverse sources. Consequently, a distinct method with a structured framework is required to avoid such complications [3–5]. One study suggested resilience primarily relies on reliability and recoverability and integrated bridge resilience with 15 parameters, broadly categorized under reliability and recovery factors [6]. While evaluating these resilience parameters is crucial in bridge resilience measurement, the analysis of the interdependencies among the parameters is also essential in achieving a robust bridge resilience system.

According to the available literature, significant research on transportation network system resiliency analysis has been undertaken. Some studies have evaluated the resilience of bridge infrastructure against natural hazards such as earthquakes and floods. For instance, Nasiopoulos et al. [7], Zhang and Wei [8], and Argyroudis et al. [9] used realistic

fragility functions and realistic restoration functions to assess the seismic resilience of bridge infrastructure. In other research, Patel et al. [10] employed multi-criteria decision-making tools such as the analytic hierarchy process (AHP) and technique for order of preference by similarity to ideal solution (TOPSIS) to evaluate bridge resilience but they only examined flood hazards. However, the primary deficiency of these studies was that they did not account for parameter dependencies, which are crucial for effectiveness. In earlier research, resilience was investigated intensively based on information about individual parameters. Hence, the primary purpose of the present novel research is to examine the interplay among various bridge infrastructure resilience parameters and seismic hazards.

Several decision-making methodologies have been developed and proposed to analyze the interdependencies of the different criteria of a specific system. Fault tree analysis (FTA), structural equation model (SEM) [11], total interpretative structural/modelling (TISM) [12], and decision-making trial and evaluation laboratory (DEMATEL) [13] are examples of tools that can be applied to evaluate the interdependencies among the influencing parameters.

Sen et al. [14] depicted that FTA offers information on inter-dependency without considering the nonlinear effect, a shortcoming of this method. The SEM has the disadvantage of requiring long questions to create dependency, while ISM and DEMATEL have the benefit of creating dependencies based on expert knowledge. In ISM, subjective data are used to build the first direct impact matrix (i.e., testing whether or not two elements affect each other). However, in DEMATEL, numerical data indicate the interdependence or impact among the parameters.

This study by Sen et al. [14] used the DEMATEL approach since this methodology uses numeric values when measuring how different parameters of bridge resilience are related. The DEMATEL approach can be integrated with other theories such as crisp, rough or fuzzy. However, the main disadvantage of the crisp DEMATEL approach is that it does not account for the vagueness perceived in experts' criteria while the methodology is developed, yielding potentially imprecise results and, therefore, wrong conclusions. In order to enhance this approach, rough theory can improve the reliability based on the conclusions obtained as it establishes lower and upper ranges through the development [15].

Therefore, the primary aims of the study are the following:

(i)    To assess the interplay among the resilience parameters of bridge infrastructure against any seismic event
(ii)    To find the effectiveness of the rough DEMATEL technique
(iii)    To compare the rough and crisp DEMATEL approaches

## 2. Rough DEMATEL Method

In this study, the interactions between bridge infrastructure resilience factors with a seismic hazard are first evaluated using the rough DEMATEL method. In this method, the crisp MEMATEL method is integrated with rough set theory to address the uncertainty, and the result is checked with that of crisp DEMATEL. The development of rough DEMATEL is described in this section, while the crisp DEMATEL method is summarized in Appendix B.

Initially, $m$ decision-makers develop $m$ direct relation matrices $Dl_{ij}$ based on the influencing ranges already established in crisp DEMATEL. Formula (1) denotes the matrix $Dl_{ij}$ of the $k$th decision-maker, where $dl_{ij}$ is the coefficient of matrix $Dl_{ij}$ and, $i, j = 1, 2, \ldots, n$.

$$D_{ij}^{rk} = \begin{bmatrix} d_{11}^{rk} & d_{12}^{rk} & \cdots & d_{1n}^{rk} \\ d_{21}^{rk} & d_{22}^{rk} & \cdots & d_{2n}^{rk} \\ \vdots & \vdots & \ddots & \vdots \\ d_{n1}^{rk} & d_{n2}^{rk} & \cdots & d_{nn}^{rk} \end{bmatrix} \tag{1}$$

Each step associated with the rough DEMATEL method development is detailed as follows [14,15]:

Step 1: Based on expert-defined matrices $D\prime_{ij}$, the direct relation matrix $\widetilde{D}_{ij}$ is developed. Different from crisp DEMATEL, this approach does not apply the average from all experts' matrices to develop matrix $\widetilde{D}_{ij}$, as shown in Equation (2):

$$\widetilde{D}_{ij} = \begin{bmatrix} 1 & \widetilde{d}_{12}^2 & \cdots & \widetilde{d}_{1n}^n \\ \widetilde{d}_{21}^1 & 1 & \cdots & \widetilde{d}_{2n}^n \\ \vdots & \vdots & \ddots & \vdots \\ \widetilde{d}_{n1}^1 & \widetilde{d}_{n2}^2 & \cdots & 1 \end{bmatrix} \tag{2}$$

where $\widetilde{d}_{ij}$ is an element of the $\widetilde{D}_{ij}$ matrix.

Step 2: The $\widetilde{D}_{ij}$ matrix elements are transformed into rough numbers ($RN$) by calculating a lower and upper approximation, as shown in Equation (3).

$$\begin{cases} Lower\ approximation,\ \underline{App}\left(d_{ij}^{rk}\right) = \cup \left\{ X \in U/J(X) \leq d_{ij}^{rk} \right\} \\ Upper\ approximation,\ \overline{App}\left(d_{ij}^{rk}\right) = \cup \left\{ X \in U/J(X) \geq d_{ij}^{rk} \right\} \end{cases} \tag{3}$$

where $U$ represents all the numbers, $X$ is an arbitrary object of $U$, $J = \left\{ d_{ij}^{r1}, d_{ij}^{r2}, \ldots, d_{ij}^{rm} \right\}$ for $m$ decision makers' opinion set. Upper and lower limits are calculated to construct the matrix $\widetilde{D}_{ij}$, as shown in Equation (4):

$$\begin{cases} \underline{Lim}\left(d_{ij}^{rk}\right) = \frac{\sum_{m=1}^{N_{tL}} X_{ij}}{N_{ijL}} \\ \overline{Lim}\left(d_{ij}^{rk}\right) = \frac{\sum_{m=1}^{N_{tU}} Y_{ij}}{N_{ijU}} \end{cases} \tag{4}$$

where $X_{ij}$ and $Y_{ij}$ represent the lower and upper approximations from $d_{ij}^{rk}$ element, respectively, and $N_{ijL}$ and $N_{ijU}$ represent the number of elements considered to calculate lower and upper approximations, respectively. Once the lower and upper limits from all elements are determined, the $\widetilde{D}_{ij}$ matrix is transformed into $RN$, as demonstrated by Formula (5):

$$RN\left(d_{ij}^{rk}\right) = \frac{\left[ \underline{Lim}\left(d_{ij}^{rk}\right), \overline{Lim}\left(d_{ij}^{rk}\right) \right]}{\left[ d_{ij}^{rKL}, d_{ij}^{rKU} \right]} \tag{5}$$

A rough number series may be established for every coefficient, as this interval represents the degree of vagueness. This sequence is shown in Equation (6):

$$RN\left(\widetilde{d}_{ij}\right) = \left\{ \left[ d_{ij}^{1L}, d_{ij}^{1U} \right], \left[ d_{ij}^{2L}, d_{ij}^{2U} \right], \ldots, \left[ d_{ij}^{mL}, d_{ij}^{mU} \right] \right\} \tag{6}$$

The average of rough intervals needs to be calculated to construct the rough group direct relation matrix $\overline{R}$. Equations (7)–(9) show the steps that must be followed to compute the rough range for $m$ decision-makers:

$$\overline{RN\left(\widetilde{d}_{ij}\right)} = \left[ d_{ij}^{-L}, d_{ij}^{-U} \right] \tag{7}$$

$$d_{ij}^{-L} = \left( \sum_{k=1}^{m} d_{ij}^{rkL} \right)/m \tag{8}$$

$$d_{ij}^{-U} = \left( \sum_{k=1}^{m} d_{ij}^{rkU} \right)/m \tag{9}$$

where $d_{ij}^{-L}$ and $d_{ij}^{-U}$ are the lower and upper limits of *RN*. With this, the matrix $\overline{R}$ is expressed by the following equation:

$$\overline{R} = \overline{[RN(\tilde{d}_{ij})]}_{nxm} = \begin{bmatrix} [1,1] & \left[d_{12}^{-L}, d_{12}^{-U}\right] & \cdots & \left[d_{1n}^{-L}, d_{1n}^{-U}\right] \\ \left[d_{21}^{-L}, d_{21}^{-U}\right] & [1,1] & \cdots & \left[d_{2n}^{-L}, d_{2n}^{-U}\right] \\ \vdots & \vdots & \ddots & \vdots \\ \left[d_{n1}^{-L}, d_{n1}^{-U}\right] & \left[d_{n2}^{-L}, d_{n2}^{-U}\right] & \cdots & [1,1] \end{bmatrix} \tag{10}$$

Step 3: Based on the rough total, the relation matrix $\overline{R}_T$ is created. The $\overline{R}$ matrix is normalized using Formula (11).

$$\overline{R'} = \overline{[RN(\tilde{d}_{ij})']}_{n \times n} = \begin{bmatrix} \overline{RN(\tilde{d}_{11})'} & \overline{RN(\tilde{d}_{12})'} & \cdots & \overline{RN(\tilde{d}_{1n})'} \\ \overline{RN(\tilde{d}_{21})'} & \overline{RN(\tilde{d}_{22})'} & \cdots & \overline{RN(\tilde{d}_{2n})'} \\ \vdots & \vdots & \ddots & \vdots \\ \overline{RN(\tilde{d}_{n1})'} & \overline{RN(\tilde{d}_{n2})'} & \cdots & \overline{RN(\tilde{d}_{nn})'} \end{bmatrix} \tag{11}$$

where $\overline{RN(\tilde{d}_{ij})'} = \dfrac{\overline{RN(\tilde{d}_{ij})}}{\tau} = \left[\dfrac{\overline{d}_{ij}^L}{\tau}, \dfrac{\overline{d}_{ij}^U}{\tau}\right]$ and $\tau = \max\limits_{1 \leq i \leq n} (\sum\limits_{j=1}^{n} \overline{d}_{ij}^U)$.

Thus, $\overline{R}_T$ matrix is expressed by the following:

$$\overline{R_T} = [r_{t_{ij}}]_{n \times m} \tag{12}$$

$$r_{t_{ij}} = [r_{t_{ij}}^L, r_{t_{ij}}^U] \tag{13}$$

$$\overline{R_T}^S = [r_{ij}^S]_{n \times n} = \overline{R'}(I - \overline{R'})^{-1}, S = L, U. \tag{14}$$

where $r_{t_{ij}}^L$ and $r_{t_{ij}}^U$ are the lower and upper limits of the rough range $r_{t_{ij}}$ in the matrix $\overline{R}_T$, respectively.

Step 4: Prominence $P_R$ and relation $R_E$ can be computed for every factor. The summations of rows $S_R$ and columns $S_C$ are obtained from matrix $\overline{R}_T$, as expressed in Formula (15):

$$\begin{aligned} S_{R_i} &= [s_{r_i}^L, s_{r_i}^U] = \left[\sum\limits_{j=1}^{n} r_{t_{ij}}^L, \sum\limits_{j=1}^{n} r_{t_{ij}}^U\right] \\ S_{C_j} &= [s_{c_j}^L, s_{c_j}^U] = \left[\sum\limits_{i=1}^{n} r_{t_{ij}}^L, \sum\limits_{i=1}^{n} r_{t_{ij}}^U\right] \end{aligned} \tag{15}$$

where $S_{r_i}^L, S_{r_i}^U$ and $S_{C_j}^L, S_{C_j}^U$ are the lower and upper limits of $S_{Ri}$ and $S_{Cj}$, respectively. Consecutively, $S_R$ and $S_C$ are transformed into crisp numbers to calculate $P_R$ and $R_E$. This transformation is accomplished by the following three phases:

*Step A*: Apply Equation (16) to normalize $S_{r_i}^L$ and $S_{r_i}^U$:

$$\begin{aligned} \tilde{s}_{r_i}^L &= (s_{r_i}^L - \min\limits_{i} s_{r_i}^L)/\Theta_{\min}^{\max} \\ \tilde{s}_{r_i}^U &= (s_{r_i}^U - \min\limits_{i} s_{r_i}^L)/\Theta_{\min}^{\max} \end{aligned} \tag{16}$$

where $\Theta_{\min}^{\max} = \max\limits_{i} s_{r_i}^U - \min\limits_{i} s_{r_i}^L$ and $\tilde{s}_{r_i}^L, \tilde{s}_{r_i}^U$ are normalized $S_{r_i}^L$ and $S_{r_i}^U$.

*Step B*: Obtain the total normalized crisp value $\sigma_i$:

$$\sigma_i = \frac{\tilde{s}_{r_i}^L \times (1 - \tilde{s}_{r_i}^L) + \tilde{s}_{r_i}^U \times \tilde{s}_{r_i}^U}{(1 - \tilde{s}_{r_i}^L + \tilde{s}_{r_i}^U)} \tag{17}$$

*Step C*: Compute the crisp value from $s_{r_i}$ for $S_{R_i}$:

$$s_{r_r} = \min_i s_{r_i}^L + \sigma_i \times \Theta_{\min}^{\max} \tag{18}$$

The final crisp value of $s_{c_j}$ for $S_{C_j}$ can be computed similarly. Finally, using Equation (19), $P_R$ and $R_E$ are evaluated:

$$\begin{aligned} P_R &= S_{r_i} + S_{c_j}, \; i = j \\ R_E &= S_{r_i} - S_{c_j}, \; i = j \end{aligned} \tag{19}$$

In order to calculate every factor's relevance, $P_R$ needs to be evaluated, where the larger the $P_R$ for a factor, the greater its impact. On the other hand, $R_E$ is used to classify the factors into cause or effect, where a positive $R_E$ is defines a cause, and a negative $R_E$ implies an effect. Finally, a causal diagram is obtained by delineating these two values ($P_R$ and $R_E$).

Step 5: A threshold value for rough approach ($\lambda'$) is calculated to build the relationship diagram. First, a crisp value for every element from the matrix $\overline{R}_T$ is obtained by Equation (A6). After assigning the $\lambda'$ value and applying the same conditions discussed in Equation (A7), the reachability matrix for the rough method is built.

## 3. Case Study

In this section, the crisp and rough DEMATEL approaches are employed to assess the interactions among the main parameters that affect the bridge infrastructure resilience based on the feedback from experts in this area. In this study, the causal diagrams illustrate the parameters and relations that are essential to be considered by stakeholders and decision-makers. Finally, the rough DEMATEL results are compared with crisp DEMATEL results to demonstrate the effectiveness of these methods.

### 3.1. Factor Selection

The most significant factors that affect the resilience of bridge structures against natural disasters are identified based on a thorough literature review (Table A1). Fifteen metrics were chosen for the two primary areas of bridge resilience: reliability and recovery. Table 1 describes the appropriateness of the metrics under reliability and recovery aspects of resilience.

**Table 1.** Resilience influencing parameters of bridge infrastructure [16].

| | Parameters | Description | Reference |
|---|---|---|---|
| Reliability | Foundation (REL 1) Abutment (REL 2) Bearing (REL 3) Piers (REL 4) Girder (REL 5) In-span joint (REL 6) | These are the major structural components that are individually and in combination responsible for achieving the reliability of earthquake-resilient bridge infrastructure. | [17–19] |
| | Age (REL 7) | Since a newly constructed structure performs better than an aged structure, the age of the structure is crucial for structural reliability. | [17,20] |
| | Design Period (REL 8) | The design philosophy of the bridges has changed over time in terms of seismic considerations. | [21,22] |
| | Bridge geometry (REL 9) | Bridge geometry is a significant component of earthquake resilience because particular designs, such as skewed bridges, are more sensitive than straight bridges regarding seismic resilience. | [23,24] |
| Recovery | Structural Monitoring (REC 1) | Constant monitoring is essential to continuously provide the status of bridges and thus determine complications at the proper time and perform quick recovery measures. | [25,26] |
| | Maintenance (REC 2) | Regular and proper maintenance programs make a bridge more capable of securing speedy recuperation. | [26] |
| | Degree of damage (REC 3) | The pace of recovery is proportional to the degree of damage since the greater the extent of damage, the slower the recovery pace. | [27] |
| | Structural Importance (REC 4) | A significant determinant for decision-makers is whether spending resources on bridge rehabilitation is worthwhile. | [25] |
| | Availability of Resources (REC 5) | The availability of construction materials has a significant impact on the pace of recovery. | [14] |
| | Approachability (REC 6) | Bridge infrastructure recovery requires special tools and instruments; hence, obstacles in accessing the location affect the recovery process. | [14] |

Before both approaches are developed, the expert opinions are presented based on their respective fields, along with the final conclusions obtained from their research. In this case study, five experts were consulted to examine the interplay between bridge infrastructure resilience elements to produce credible conclusions. Each specialist has ten to thirty years of professional experience in bridge design, management, and operation (Table 2).

**Table 2.** Details of the Experts.

| Decision Makers | Profile | Experience (Years) | Roll in Work/Designation |
|---|---|---|---|
| DM 1 | PhD. P. Eng. | 11 | Assistant Professor, Structural Engineering Dept. and Former Bridge Design Engineer |
| DM 2 | MSc. P. Eng. | 31 | Senior Principal Engineer (Transport infrastructure) |
| DM 3 | MSc. | 11 | Bridge Project Manager, Govt. Transport Institution |
| DM 4 | MSc. | 10 | Bridge Project Manager, Govt. Transport Institution |
| DM 5 | MSc. | 10 | Researcher, Bridge Engineering and Former Bridge Design Engineer, Govt. Transport Institution |

### 3.2. Causal and Relationship Diagram by Crisp DEMATEL

In order to perform crisp DEMATEL, our five decision-makers expressed the relationship among these reliability and recovery factors using the following scale: 1—no impact; 2—low impact; 3—medium impact; 4—high impact; and 5—very high impact. Tables A2 and A3 list the expert opinions on the reliability and recovery factors, respectively. Next, the average was calculated using Equation (A1), and matrices were normalized for both resilience factors, as shown in Tables A4 and A5. Using Equation (A3), total-relation matrices were developed for reliability (Table A6) and recovery (Table A7) factors. Finally, the prominence and relation values for the reliability and recovery factors were calculated using the DEMATEL method and shown in Table 3.

**Table 3.** Prominence and relation values for resilience factors using crisp DEMATEL.

| | Reliability Factors | | | | | Recovery Factors | | | |
|---|---|---|---|---|---|---|---|---|---|
| Parameter | D | R | D + R | D − R | Parameter | D | R | D + R | D − R |
| REL1 | 3.76 | 4.33 | 8.09 | −0.58 | REC1 | 9.22 | 9.75 | 18.97 | −0.52 |
| REL2 | 3.42 | 4.11 | 7.53 | −0.69 | REC2 | 10.06 | 13.15 | 23.21 | −3.08 |
| REL3 | 3.55 | 4.39 | 7.94 | −0.84 | REC3 | 9.46 | 10.65 | 20.11 | −1.19 |
| REL4 | 4.00 | 4.59 | 8.59 | −0.58 | REC4 | 10.65 | 8.73 | 19.38 | 1.92 |
| REL5 | 4.13 | 4.55 | 8.68 | −0.41 | REC5 | 10.85 | 9.19 | 20.04 | 1.66 |
| REL6 | 2.93 | 4.42 | 7.36 | −1.49 | REC6 | 9.62 | 8.39 | 18.01 | 1.23 |
| REL7 | 4.75 | 1.74 | 6.49 | 3.01 | | | | | |
| REL8 | 4.17 | 2.02 | 6.20 | 2.15 | | | | | |
| REL9 | 3.55 | 4.11 | 7.66 | −0.57 | | | | | |

Figure 1a,b depict the resulting causal diagrams for reliability and recovery, which were constructed using Equations (A6) and (A7). Using Equation (A6), the threshold value $\lambda$ for dependability is 0.4231, while the threshold value for recovery is 1.6626. For a better understanding, the relationship is divided into three categories: no influence, moderate influence (ga), and high influence (red). In the case of reliability, if $r_t < 0.4230$, then no influence exists in the relation; for the condition $0.4230 < r_t < 0.54$, the relationship is classified as a moderate influence, and a high influence occurs when $r_t > 0.54$. Likewise, in the case of recovery, when $r_t < 1.6626$, the relation has no influence; when $1.6626 < r_t < 2$,

the relationship is defined as moderate influence, and for the relationship to have a high influence, $r_t > 2$. These relations are depicted in Figure 2a,b.

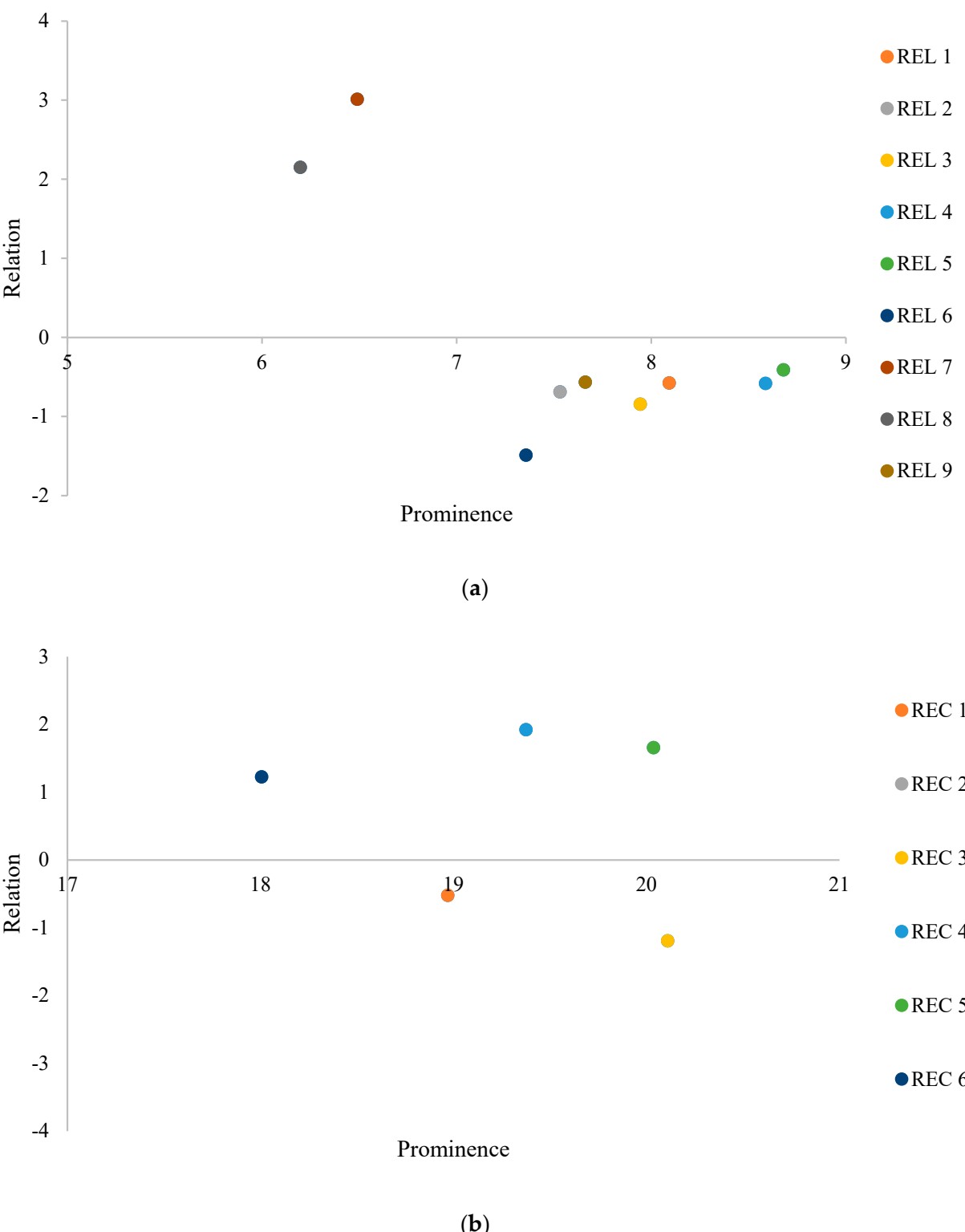

**Figure 1.** Causal diagram for (**a**) reliability and (**b**) recovery for crisp DEMATEL.

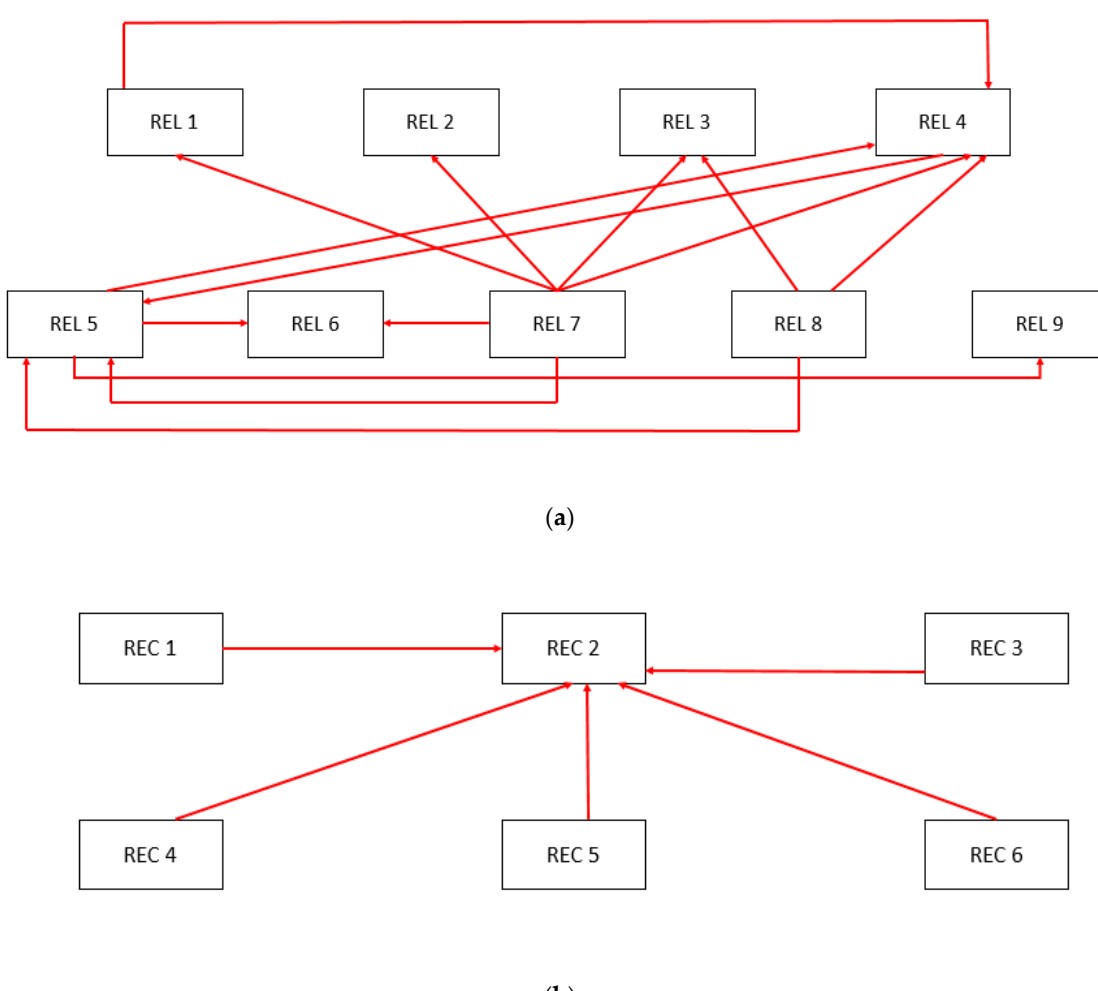

**Figure 2.** Relationship diagram for (**a**) reliability and (**b**) recovery for crisp DEMATEL.

### 3.3. Causal and Relationship Diagram by Rough DEMATEL

In order to perform a rough DEMATEL analysis, the same expert opinions (Tables A2 and A3) are considered. The normalized rough direct-relation matrices were calculated for reliability (Table A8) and recovery (Table A9) factors using Equations (3)–(11). Then, the rough total-relation matrices for reliability and recovery factors were determined using Equations (12)–(14), as shown in Tables A10 and A11, respectively. Finally, the prominence and relation values for the reliability and recovery factors were produced using Equations (15)–(19), as highlighted in Table 4. Figure 3a,b show the resulting causal diagrams for reliability and recovery using the rough DEMATEL method.

**Table 4.** Prominence and relation values for resilience factors using rough DEMATEL.

| | Reliability Factors | | | | | Recovery Factors | | | |
| --- | --- | --- | --- | --- | --- | --- | --- | --- | --- |
| Parameter | D | R | D + R | D − R | Parameter | D | R | D + R | D − R |
| REL1 | 2.83 | 3.44 | 6.27 | −0.61 | REC1 | 6.94 | 7.19 | 14.12 | −0.25 |
| REL2 | 2.46 | 3.24 | 5.70 | −0.77 | REC2 | 8.00 | 10.03 | 18.03 | −2.04 |
| REL3 | 2.54 | 3.54 | 6.08 | −1.01 | REC3 | 7.38 | 7.91 | 15.28 | −0.53 |
| REL4 | 3.00 | 3.75 | 6.75 | −0.75 | REC4 | 8.48 | 6.25 | 14.72 | 2.23 |
| REL5 | 3.12 | 3.78 | 6.90 | −0.66 | REC5 | 8.36 | 6.82 | 15.18 | 1.54 |
| REL6 | 1.96 | 3.58 | 5.54 | −1.62 | REC6 | 7.23 | 6.11 | 13.34 | 1.12 |
| REL7 | 3.83 | 1.01 | 4.84 | 2.82 | | | | | |
| REL8 | 3.29 | 1.34 | 4.63 | 1.96 | | | | | |
| REL9 | 2.59 | 4.20 | 6.78 | −1.61 | | | | | |

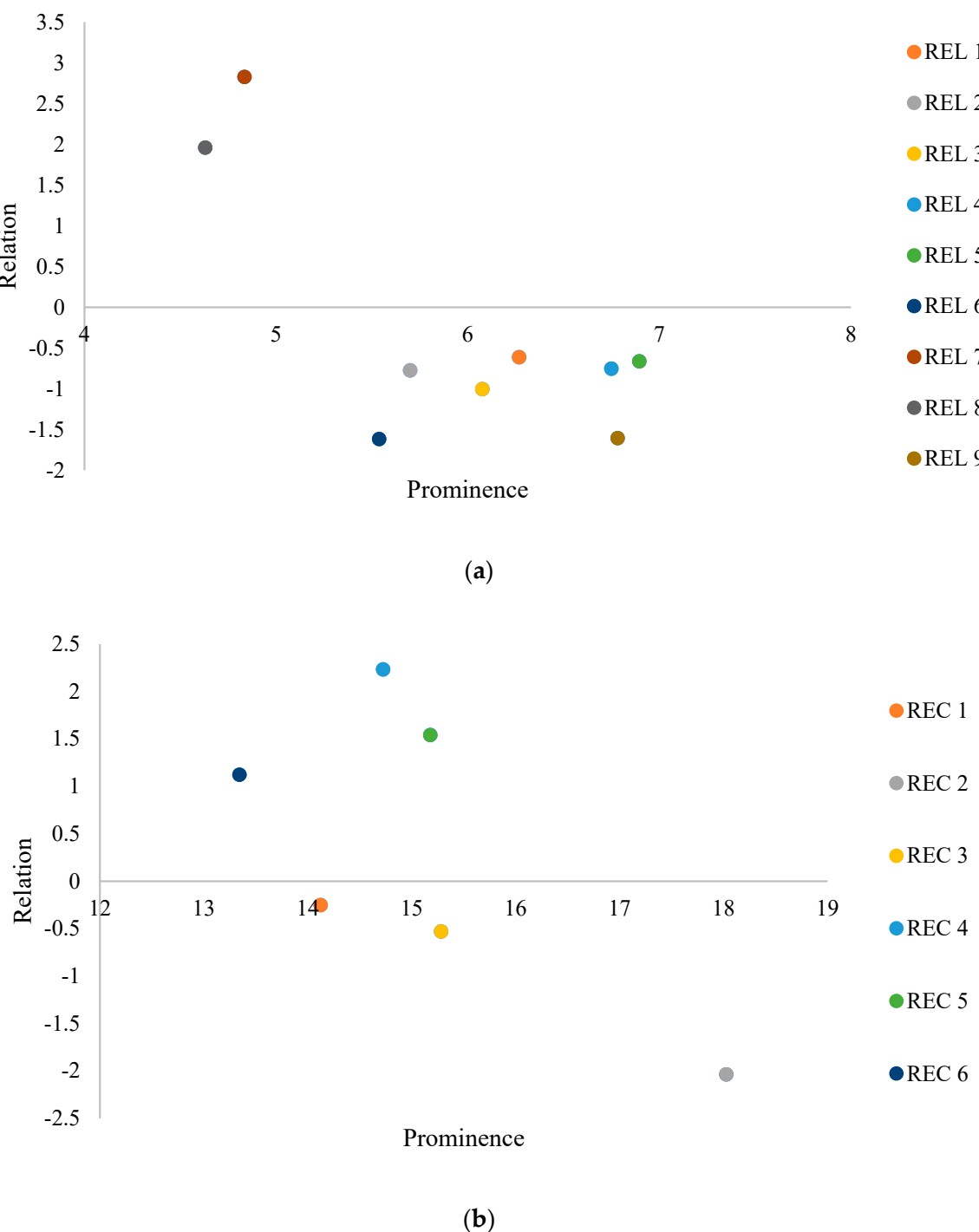

**Figure 3.** Causal diagram for (**a**) reliability and (**b**) recovery for rough DEMATEL.

For a better understanding, the relationship is divided into the same three categories: no influence, moderate influence (grey), and high influence (red). In the case of reliability, if $r_t < 0.3259$, then no influence is in the relation; for the condition $0.3259 < r_t < 0.42$, the relationship is classified as a moderate influence, and a relationship exhibits a high influence when $r_t > 0.42$. Likewise, in the case of recovery, when $r_t < 1.2299$, the relation has no influence when $1.2299 < r_t < 1.45$, the relationship is defined as a moderate influence, and for a high influence relation, $r_t > 1.45$. The relations are depicted in Figure 4a,b.

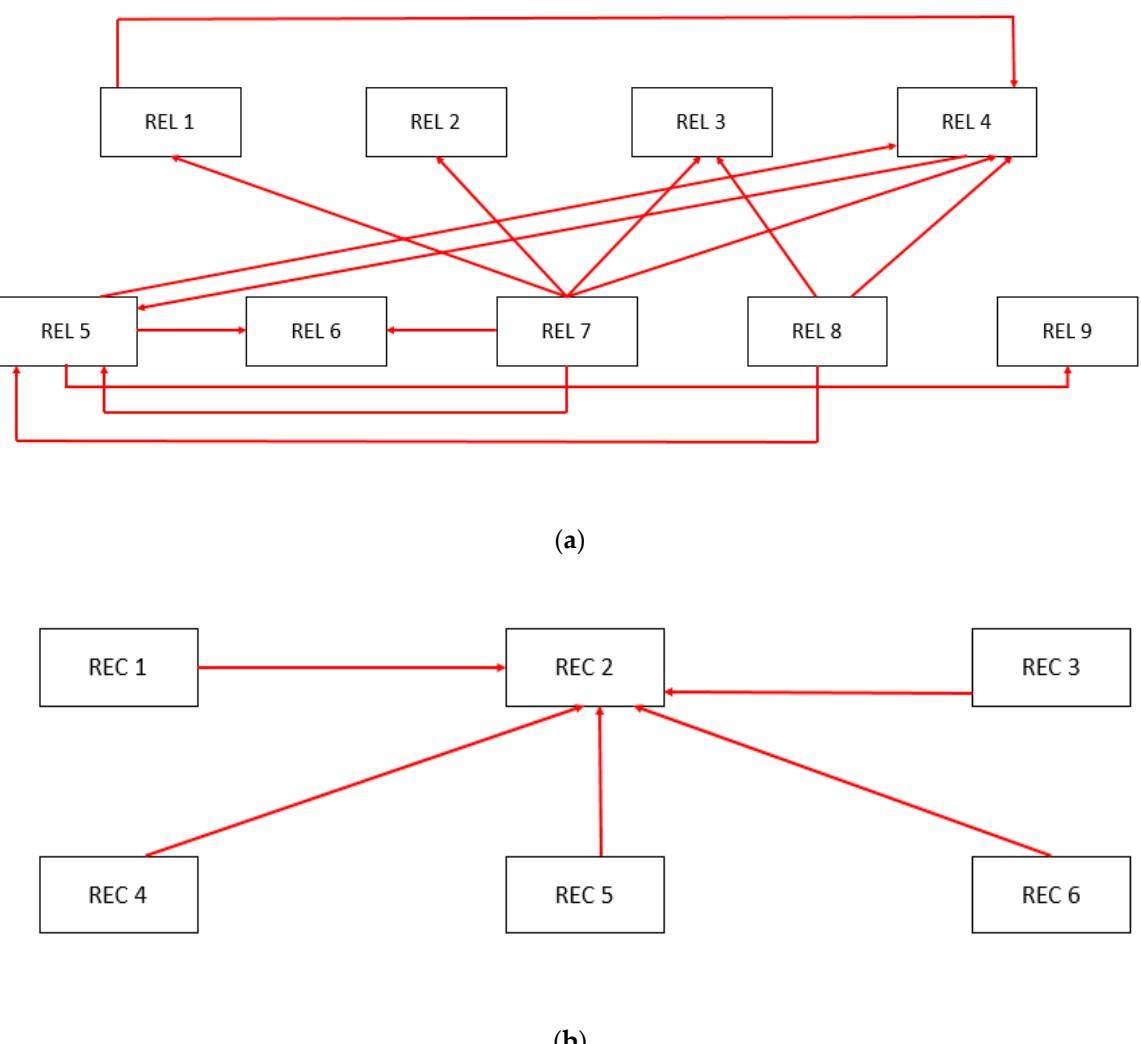

**Figure 4.** Relationship diagram for (**a**) reliability and (**b**) recovery for rough DEMATEL.

## 4. Results and Discussion

Figures 1a and 3a show that by using both approaches, the most critical parameter for reliability is REL 5 (*girder*). The main difference between these causal diagrams is that the factor REL 9 (*bridge geometry*) is more significant than parameters REL 3 (*bearing*), REL 1 (*foundation*), and REL 4 (*piers*) with crisp DEMATEL, while parameters REL 3, REL 1, and REL 4 are more important than factor REL 9 with rough DEMATEL. Similarly, Figures 1b and 3b show that the most critical parameter for recovery is REC 2 (*maintenance*). Both approaches show the same order of prominence among the parameters, but minimal differences between REC 4 (*structural importance*), REC 1 (*structural monitoring*), REC 5 (*availability of resources*), and REC 3 (*degree of damage*) are evident in the crisp DEMATEL approach, while in the rough one, the prominence difference is large enough to establish a clear order.

Similarly, in terms of reliability, Tables 3 and 4 and Figures 1a and 3a show that, of all the factors, only *age* (REL 7) and *design period* (REL 8) are in the cause group. In both approaches, *age* (REL 7) has the greater $(D - R)$ score of 3.01 in crisp DEMATEL and 2.82 in rough DEMATEL, implying that REL 7 has more impact. Moreover, *age* (REL 7) has a greater $(D + R)$ value of 6.49 in crisp DEMATEL and 4.84 in rough DEMATEL, indicating that the reliability parameter *age* (REL 7) can enhance the bridge infrastructure resilience against any seismic event. Consequently, the assessment based on decision-makers' judgments affirms that *age* is the most influential criterium of the bridge infrastructure's total seismic

resiliency. Because six of the nine criteria of the reliability parameters are distinct key structural components of bridge infrastructure, declines in their performances are expected with time. As a corollary, in addition to the six structural element characteristics, the reliability parameter *age* directly impacts the majority of the parameters.

Likewise, Tables 3 and 4 and Figures 1b and 3b show that the *structural importance* (REC 4), *availability of resources* (REC 5), and *approachability* (REC 6) are in the cause group for recovery. *Structural importance* (REC 4) has the highest (D-R) score in both methods, with 1.92 in crisp DEMATEL and 2.23 in rough DEMATEL, suggesting that REC 4 has a more considerable effect. However, the *availability of resources* (REC 5) has the greatest (D+R) values, with 20.04 in crisp DEMATEL and 15.18 in rough DEMATEL, showing that the *availability of resources* can increase the recovery factor of bridge infrastructure resilience for seismic disasters.

Furthermore, Figures 2a and 4a show the relationship diagrams for reliability factors based on threshold values, and 14 red arrows (high influence) are present for both approaches, along with 37 grey arrows (moderate influence, not shown in the figure) with crisp DEMATEL and 35 grey arrows (not shown in the figure) with rough DEMATEL. The most crucial scenario observed from both figures is that almost all the arrows are related, with *age* (REL 7) as a cause and pier (REL 4) as a consequence.

Figures 2b and 4b show the relationship diagrams for recovery factors. Five red arrows (high influence), all related to *maintenance* (REC 2) as a consequence and seven grey arrows (moderate influence, not shown in the figure) are present in both approaches. The crisp DEMATEL method exhibits a relation between the *approachability* factor (REC6) as a cause and the *degree of damage* (REC 3) as a consequence. In rough DEMATEL, this relation does not exist. Still, a relation appears between *structural importance* (REC 4) as a cause and the *availability of resources* (REC 5) as a consequence.

Despite the results showing almost the same relations among the factors, the minimal differences found between the methodologies should be considered because the rough DE-MATEL approach offers a critical element, unlike the crisp DEMANTAL—a consideration of the vagueness in the decision-making process. This is mainly the result of the mechanism used to manipulate different judgments. To clarify with an example, consider the answers obtained from the five experts on how *foundation* (REL 1) affects *abutment* (REL 2). The ranges of influences established were {5, 4, 5, 5, 5}, which represents rough intervals of {(4.8, 5), (4, 4.8), (4.8, 5), (4.8, 5), (4.8, 5)}; as a result, the interval is {(4.64, 4.96)}. By settling this range, influences of judgment distribution are taken into consideration, providing relative certainty to make correct decisions. On the other hand, following the DEMATEL crisp method, 4.8 is the result, which does not offer the desired certainty, considering how subjectivity can affect the prominence of the relation. Therefore, rough DEMATEL should be used over crisp DEMATEL.

While Sen et al. [14] worked with housing infrastructure resilience for flood hazards, they also considered two similar criteria—*structural monitoring* and *maintenance*—along with other criteria under the recovery factor. They revealed a strong relationship between *structural monitoring* and *maintenance*. Since this is the first study that explores the relationship between the factors of bridge resilience, we cannot compare the results with the other researchers. However, we can compare studies of rough DEMATEL and crisp DEMATEL methods.

## 5. Conclusions

Identifying the interactions among the resilience criteria is critical to assessing a bridge's overall seismic resilience. This paper implements a method for evaluating the interplay among seismically resilient bridge infrastructure variables. Intensive literature research was performed to find the critical reliability and recovery parameters that reflect the bridge infrastructure's resilience. This study employed the crisp DEMATEL and rough DEMATEL techniques to survey the interplay across resilience elements. Then, causal and relationship diagrams for reliability and recovery variables were generated.

The results of the causal diagram divide the bridge infrastructure recovery and reliability variables into cause-and-effect parameters. The results revealed that *age* is the most crucial reliability factor for the cause group, whereas *piers* is the most critical parameter for the impact group. Likewise, *structural importance* and the *availability of resources* are crucial recovery parameters in the cause group, whereas *maintenance* is vital in the effect group. The study's outcome suggested that rough DEMATEL is the preferred technique since it considers the ambiguity of expert opinions. The DEMATEL strategy is resilient and flexible for all asset managers facing difficulties requiring collaborative decision-making in an uncertain setting. For this reason, the DEMATEL technique provides decision-makers and stakeholders with more credible outcomes and data.

The following are some of the significant advances made by this study in the domain of bridge infrastructure resilience against seismic hazards:

- Crucial bridge resilience parameters and their interrelationships were identified.
- The bridge resilience indexes were evaluated by comparing the results from two well-established multi-criteria decision-making tools.
- Assistance to the stakeholders and policymakers was provided for preparing for future unforeseeable scenarios and hazards from the determinations of the study.

This study focuses on earthquake hazards. In the future, it can be augmented by analyses of other natural disasters, such as landslides, floods, avalanches, and tsunamis, that impact bridge infrastructure resilience. In addition, other parameters can be included to develop a more comprehensive framework for resilience assessment. Finally, both crisp and rough DEMATEL techniques do not distinguish between positive and negative effects. This may be examined further to enhance this method and offer a more authentic assessment of resilience factors.

**Author Contributions:** Conceptualization, Á.F.G.R., M.S.A.K., G.K., M.B. and S.D.; methodology, Á.F.G.R. and M.S.A.K.; software, Á.F.G.R. and M.S.A.K.; validation, M.K, G.K., M.B. and S.D.; formal analysis, Á.F.G.R. and M.S.A.K.; investigation, Á.F.G.R. and M.S.A.K.; resources, G.K., M.B. and S.D.; data curation, Á.F.G.R. and M.S.A.K.; writing—original draft preparation, Á.F.G.R. and M.S.A.K.; writing—review and editing, G.K., M.B. and S.D.; visualization, Á.F.G.R. and M.S.A.K.; supervision, G.K.; project administration, G.K., M.B. and S.D. All authors have read and agreed to the published version of the manuscript.

**Funding:** The authors acknowledge the financial support through the Mitacs Globalink Research Internship program, Canada.

**Institutional Review Board Statement:** Not applicable.

**Informed Consent Statement:** Not applicable.

**Data Availability Statement:** Some or all data, models, or code that support the findings of this study are available from the corresponding author upon reasonable request.

**Acknowledgments:** The authors would like to thank the experts in providing their feedback for performing this study.

**Conflicts of Interest:** The authors declare no conflict of interest.

**Appendix A.**

**Table A1.** Major factors of physical infrastructure resilience against natural hazards [16].

| Topic of the Study | Factors of Resilience | Reference |
|---|---|---|
| Resilience assessment of housing infrastructure against flood | Reliability<br>Recovery<br>Resistance | [14] |
| Resilience assessment of urban transportation systems | Re-building of critical functionality<br>Re-stabilization of critical functionality<br>Reconfiguration after recovery | [28] |

**Table A1.** *Cont.*

| Topic of the Study | Factors of Resilience | Reference |
|---|---|---|
| Seismic resilience assessment using fuzzy sets theory | Target functionality<br>Residual functionality<br>Recovery time<br>Idle time interval | [29] |
| Structural resilience against natural hazards | Preparedness<br>Management<br>Risk analysis | [26] |
| Review paper on civil infrastructure resilience | Risk assessment<br>Disasters risk management<br>Water supply<br>Water resources<br>Climate change<br>Economic and social resources<br>Strategic planning<br>Decision making<br>Sustainable development | [25] |
| Resilience assessment of single system to interdependent systems | Damage propagation<br>Disasters prevention<br>Assessment and recovery | [30] |

**Table A2.** $\widetilde{D}_{ij}$ Matrix for reliability.

|      | REL1 | REL2 | REL3 | REL4 | REL5 | REL6 | REL7 | REL8 | REL9 |
|------|------|------|------|------|------|------|------|------|------|
| REL1 | 1,1,1,1,1 | 5,4,5,5,5 | 2,2,3,4,2 | 4,4,4,5,5 | 1,4,2,4,1 | 2,4,2,4,2 | 1,1,1,1,1 | 3,1,3,1,1 | 3,5,1,1,1 |
| REL2 | 4,4,4,2,4 | 1,1,1,1,1 | 2,2,4,3,3 | 2,3,2,4,2 | 2,4,2,4,2 | 1,3,3,4,2 | 1,1,1,1,1 | 2,1,2,1,1 | 4,4,1,2,4 |
| REL3 | 3,2,3,3,2 | 2,2,2,4,2 | 1,1,1,1,1 | 4,2,4,4,3 | 3,4,3,5,4 | 3,4,3,4,3 | 2,1,1,1,1 | 1,1,1,1,1 | 3,3,3,2,3 |
| REL4 | 4,4,5,4,4 | 2,2,1,4,2 | 4,3,4,4,3 | 1,1,1,1,1 | 3,4,5,4,5 | 3,4,2,4,4 | 1,1,1,1,1 | 1,1,2,1,1 | 4,4,3,4,4 |
| REL5 | 4,4,2,4,3 | 4,4,3,4,3 | 3,3,4,4,3 | 4,4,2,4,4 | 1,1,1,1,1 | 4,4,4,5,4 | 1,1,1,1,1 | 1,1,1,1,1 | 5,5,4,4,5 |
| REL6 | 2,2,2,1,2 | 3,2,1,1,3 | 2,4,2,2,2 | 3,2,1,3,3 | 3,3,1,5,3 | 1,1,1,1,1 | 2,1,1,1,1 | 1,1,1,1,1 | 3,3,4,1,3 |
| REL7 | 4,4,5,2,4 | 3,3,3,2,4 | 5,5,5,4,5 | 5,4,5,2,4 | 4,4,5,3,4 | 3,4,3,3,4 | 1,1,1,1,1 | 4,4,4,1,4 | 1,3,1,1,1 |
| REL8 | 4,4,4,1,4 | 4,3,3,1,4 | 5,3,5,1,5 | 5,5,4,1,5 | 3,3,5,1,4 | 3,3,4,1,3 | 2,3,3,1,2 | 1,1,1,1,1 | 1,3,1,1,3 |
| REL9 | 3,3,3,1,4 | 3,3,3,1,4 | 4,3,4,1,4 | 3,4,3,1,4 | 4,4,4,1,4 | 4,2,4,1,4 | 1,1,1,1,2 | 1,1,1,1,2 | 1,1,1,1,1 |

**Table A3.** $\widetilde{D}_{ij}$ Matrix for recovery.

|      | REC1 | REC2 | REC3 | REC4 | REC5 | REC6 |
|------|------|------|------|------|------|------|
| REC1 | 1,1,1,1,1 | 4,4,4,4,5 | 4,4,4,4,4 | 2,2,4,4,2 | 1,1,1,5,1 | 1,1,1,5,1 |
| REC2 | 1,1,1,4,2 | 1,1,1,1,1 | 5,4,5,4,5 | 4,4,4,4,2 | 3,2,4,5,2 | 2,2,3,5,2 |
| REC3 | 3,3,4,4,2 | 5,5,5,4,5 | 1,1,1,1,1 | 1,1,1,4,1 | 3,3,3,5,1 | 1,1,4,5,1 |
| REC4 | 2,4,3,4,4 | 4,4,4,4,4 | 5,2,2,4,5 | 1,1,1,1,1 | 3,3,2,5,3 | 3,3,1,5,3 |
| REC5 | 4,4,4,5,4 | 5,5,5,5,5 | 2,2,2,5,2 | 2,3,2,5,2 | 1,1,1,1,1 | 3,3,3,3,3 |
| REC6 | 3,3,3,5,3 | 4,4,4,5,4 | 1,1,1,5,1 | 2,2,2,5,2 | 3,3,4,3,3 | 1,1,1,1,1 |

**Table A4.** Normalized direct relation matrix for reliability.

|      | REL1 | REL2 | REL3 | REL4 | REL5 | REL6 | REL7 | REL8 | REL9 |
|------|------|------|------|------|------|------|------|------|------|
| REL1 | 0.035 | 0.167 | 0.090 | 0.153 | 0.083 | 0.097 | 0.035 | 0.063 | 0.076 |
| REL2 | 0.125 | 0.035 | 0.097 | 0.090 | 0.097 | 0.090 | 0.035 | 0.049 | 0.104 |
| REL3 | 0.090 | 0.083 | 0.035 | 0.118 | 0.132 | 0.118 | 0.042 | 0.035 | 0.097 |
| REL4 | 0.146 | 0.076 | 0.125 | 0.035 | 0.146 | 0.118 | 0.035 | 0.042 | 0.132 |
| REL5 | 0.118 | 0.125 | 0.118 | 0.125 | 0.035 | 0.146 | 0.035 | 0.035 | 0.160 |
| REL6 | 0.063 | 0.069 | 0.083 | 0.083 | 0.104 | 0.035 | 0.042 | 0.035 | 0.097 |
| REL7 | 0.132 | 0.104 | 0.167 | 0.139 | 0.139 | 0.118 | 0.035 | 0.118 | 0.049 |
| REL8 | 0.118 | 0.104 | 0.132 | 0.139 | 0.111 | 0.097 | 0.076 | 0.035 | 0.063 |
| REL9 | 0.097 | 0.097 | 0.111 | 0.104 | 0.118 | 0.104 | 0.042 | 0.042 | 0.035 |

**Table A5.** Normalized direct relation matrix for recovery.

|  | REC1 | REC2 | REC3 | REC4 | REC5 | REC6 |
|---|---|---|---|---|---|---|
| REC1 | 0.054 | 0.226 | 0.215 | 0.151 | 0.097 | 0.097 |
| REC2 | 0.097 | 0.054 | 0.247 | 0.194 | 0.172 | 0.151 |
| REC3 | 0.172 | 0.258 | 0.054 | 0.086 | 0.161 | 0.129 |
| REC4 | 0.183 | 0.215 | 0.194 | 0.054 | 0.172 | 0.161 |
| REC5 | 0.226 | 0.269 | 0.140 | 0.151 | 0.054 | 0.161 |
| REC6 | 0.183 | 0.226 | 0.097 | 0.140 | 0.172 | 0.054 |

**Table A6.** Total-relation matrix for reliability.

|  | REL1 | REL2 | REL3 | REL4 | REL5 | REL6 | REL7 | REL8 | REL9 |
|---|---|---|---|---|---|---|---|---|---|
| REL1 | 0.415 | 0.513 | 0.468 | 0.540 | 0.478 | 0.480 | 0.184 | 0.233 | 0.440 |
| REL2 | 0.462 | 0.364 | 0.440 | 0.452 | 0.454 | 0.440 | 0.171 | 0.205 | 0.431 |
| REL3 | 0.444 | 0.420 | 0.395 | 0.488 | 0.499 | 0.479 | 0.182 | 0.197 | 0.440 |
| REL4 | 0.537 | 0.460 | 0.523 | 0.461 | 0.558 | 0.526 | 0.193 | 0.224 | 0.514 |
| REL5 | 0.526 | 0.512 | 0.529 | 0.555 | 0.472 | 0.562 | 0.199 | 0.224 | 0.551 |
| REL6 | 0.357 | 0.348 | 0.379 | 0.392 | 0.410 | 0.338 | 0.158 | 0.169 | 0.379 |
| REL7 | 0.605 | 0.555 | 0.641 | 0.639 | 0.634 | 0.604 | 0.225 | 0.334 | 0.509 |
| REL8 | 0.534 | 0.500 | 0.550 | 0.577 | 0.548 | 0.525 | 0.241 | 0.228 | 0.466 |
| REL9 | 0.450 | 0.433 | 0.465 | 0.476 | 0.487 | 0.466 | 0.182 | 0.204 | 0.380 |

**Table A7.** Total-relation matrix for recovery.

|  | REC1 | REC2 | REC3 | REC4 | REC5 | REC6 |
|---|---|---|---|---|---|---|
| REC1 | 1.421 | 2.055 | 1.704 | 1.371 | 1.393 | 1.278 |
| REC2 | 1.598 | 2.079 | 1.857 | 1.515 | 1.576 | 1.433 |
| REC3 | 1.562 | 2.128 | 1.600 | 1.352 | 1.478 | 1.335 |
| REC4 | 1.752 | 2.334 | 1.914 | 1.474 | 1.655 | 1.515 |
| REC5 | 1.814 | 2.414 | 1.907 | 1.593 | 1.576 | 1.540 |
| REC6 | 1.597 | 2.133 | 1.668 | 1.420 | 1.509 | 1.286 |

**Table A8.** Normalized rough group direct-relation matrix for reliability.

|  | REL1 | REL2 | REL3 | REL4 | REL5 | REL6 | REL7 | REL8 | REL9 |
|---|---|---|---|---|---|---|---|---|---|
| REL1 | (0.031, 0.031) | (0.145, 0.155) | (0.068, 0.096) | (0.130, 0.145) | (0.051, 0.101) | (0.072, 0.103) | (0.031, 0.031) | (0.041, 0.071) | (0.042, 0.098) |
| REL2 | (0.103, 0.123) | (0.031, 0.031) | (0.074, 0.102) | (0.068, 0.096) | (0.072, 0.103) | (0.060, 0.102) | (0.031, 0.031) | (0.036, 0.051) | (0.072, 0.116) |
| REL3 | (0.074, 0.089) | (0.065, 0.085) | (0.031, 0.031) | (0.092, 0.120) | (0.105, 0.133) | (0.099, 0.114) | (0.032, 0.042) | (0.031, 0.031) | (0.082, 0.093) |
| REL4 | (0.127, 0.137) | (0.053, 0.086) | (0.105, 0.120) | (0.031, 0.031) | (0.117, 0.145) | (0.092, 0.120) | (0.031, 0.031) | (0.032, 0.042) | (0.114, 0.124) |
| REL5 | (0.092, 0.120) | (0.105, 0.120) | (0.099, 0.114) | (0.103, 0.123) | (0.031, 0.031) | (0.127, 0.137) | (0.031, 0.031) | (0.031, 0.031) | (0.137, 0.152) |
| REL6 | (0.051, 0.061) | (0.046, 0.079) | (0.065, 0.085) | (0.060, 0.088) | (0.072, 0.116) | (0.031, 0.031) | (0.032, 0.042) | (0.031, 0.031) | (0.071, 0.104) |
| REL7 | (0.102, 0.135) | (0.083, 0.105) | (0.145, 0.155) | (0.104, 0.144) | (0.114, 0.136) | (0.099, 0.114) | (0.031, 0.031) | (0.091, 0.121) | (0.033, 0.054) |
| REL8 | (0.091, 0.121) | (0.073, 0.113) | (0.090, 0.146) | (0.097, 0.149) | (0.073, 0.127) | (0.071, 0.104) | (0.054, 0.082) | (0.031, 0.031) | (0.041, 0.071) |
| REL9 | (0.071, 0.104) | (0.071, 0.104) | (0.079, 0.119) | (0.073, 0.113) | (0.091, 0.121) | (0.072, 0.116) | (0.032, 0.042) | (0.032, 0.042) | (0.031, 0.031) |

**Table A9.** Normalized rough group direct-relation matrix for recovery.

|  | REC1 | REC2 | REC3 | REC4 | REC5 | REC6 |
|---|---|---|---|---|---|---|
| REC1 | (0.048, 0.048) | (0.195, 0.211) | (0.193, 0.193) | (0.112, 0.158) | (0.056, 0.118) | (0.056, 0.118) |
| REC2 | (0.058, 0.120) | (0.048, 0.048) | (0.211, 0.234) | (0.158, 0.189) | (0.119, 0.192) | (0.107, 0.168) |
| REC3 | (0.133, 0.176) | (0.224, 0.240) | (0.048, 0.048) | (0.054, 0.100) | (0.111, 0.179) | (0.069, 0.161) |
| REC4 | (0.142, 0.185) | (0.193, 0.193) | (0.134, 0.211) | (0.048, 0.048) | (0.130, 0.181) | (0.111, 0.179) |
| REC5 | (0.195, 0.211) | (0.242, 0.242) | (0.102, 0.149) | (0.107, 0.168) | (0.048, 0.048) | (0.145, 0.145) |
| REC6 | (0.149, 0.180) | (0.195, 0.211) | (0.056, 0.118) | (0.102, 0.149) | (0.147, 0.162) | (0.048, 0.048) |

**Table A10.** Rough total-relation matrix for reliability.

|  | REL1 | REL2 | REL3 | REL4 | REL5 | REL6 | REL7 | REL8 | REL9 |
|---|---|---|---|---|---|---|---|---|---|
| REL1 | (0.168, 0.458) | (0.260, 0.557) | (0.198, 0.528) | (0.259, 0.593) | (0.183, 0.567) | (0.201, 0.544) | (0.085, 0.196) | (0.101, 0.264) | (0.166, 0.520) |
| REL2 | (0.219, 0.502) | (0.145, 0.409) | (0.190, 0.494) | (0.191, 0.510) | (0.188, 0.526) | (0.178, 0.504) | (0.080, 0.181) | (0.090, 0.229) | (0.180, 0.497) |
| REL3 | (0.206, 0.465) | (0.187, 0.451) | (0.163, 0.420) | (0.225, 0.522) | (0.233, 0.545) | (0.229, 0.508) | (0.086, 0.188) | (0.091, 0.206) | (0.206, 0.470) |
| REL4 | (0.271, 0.552) | (0.197, 0.499) | (0.251, 0.549) | (0.189, 0.491) | (0.261, 0.606) | (0.242, 0.561) | (0.093, 0.195) | (0.101, 0.237) | (0.250, 0.544) |
| REL5 | (0.249, 0.544) | (0.248, 0.534) | (0.253, 0.550) | (0.261, 0.581) | (0.190, 0.510) | (0.279, 0.582) | (0.096, 0.198) | (0.103, 0.230) | (0.278, 0.574) |
| REL6 | (0.153, 0.393) | (0.140, 0.399) | (0.165, 0.423) | (0.164, 0.443) | (0.171, 0.477) | (0.133, 0.381) | (0.074, 0.170) | (0.077, 0.184) | (0.164, 0.432) |
| REL7 | (0.276, 0.643) | (0.244, 0.601) | (0.315, 0.675) | (0.283, 0.694) | (0.284, 0.699) | (0.273, 0.648) | (0.103, 0.232) | (0.170, 0.356) | (0.197, 0.566) |
| REL8 | (0.227, 0.603) | (0.199, 0.581) | (0.224, 0.638) | (0.236, 0.667) | (0.207, 0.659) | (0.206, 0.610) | (0.110, 0.268) | (0.094, 0.257) | (0.167, 0.555) |
| REL9 | (0.191, 0.503) | (0.182, 0.492) | (0.197, 0.526) | (0.196, 0.543) | (0.208, 0.563) | (0.192, 0.535) | (0.082, 0.198) | (0.087, 0.228) | (0.144, 0.437) |

**Table A11.** Rough total-relation matrix for recovery.

|  | REC1 | REC2 | REC3 | REC4 | REC5 | REC6 |
|---|---|---|---|---|---|---|
| REC1 | (0.309, 1.781) | (0.584, 2.319) | (0.492, 2.019) | (0.342, 1.730) | (0.295, 1.816) | (0.264, 1.703) |
| REC2 | (0.352, 2.051) | (0.489, 2.424) | (0.518, 2.256) | (0.397, 1.933) | (0.373, 2.071) | (0.329, 1.925) |
| REC3 | (0.382, 1.994) | (0.601, 2.466) | (0.356, 1.995) | (0.290, 1.776) | (0.338, 1.962) | (0.274, 1.828) |
| REC4 | (0.442, 2.174) | (0.648, 2.639) | (0.484, 2.316) | (0.321, 1.877) | (0.399, 2.130) | (0.349, 1.998) |
| REC5 | (0.515, 2.133) | (0.731, 2.602) | (0.493, 2.207) | (0.406, 1.933) | (0.348, 1.954) | (0.402, 1.916) |
| REC6 | (0.427, 1.935) | (0.616, 2.364) | (0.392, 1.995) | (0.358, 1.759) | (0.394, 1.887) | (0.276, 1.669) |

## Appendix B.

*Appendix B.1. Crisp DEMATEL Approach*

The steps in the crisp DEMATEL method follow [13,14]:

First, the different set of ranges that determine the influence among parameters is selected based on expert knowledge.

Step 1: Based on the experts' opinions, build $N$ direct-relation matrices; once they have performed the classification from factors per the interval of influence indicated. If the case study considers the criteria from different experts, build matrix $D_{ij}$ by calculating the average from all the matrices obtained. Then, $D_{ij}$ is the direct relation matrix and $d_{ij}$ is every element from the matrix, $i, j = 1, 2, \ldots, n$.

$$[D_{ij}] = \begin{bmatrix} d_{11} & d_{12} \ldots & d_{1n} \\ d_{21} & d_{22} \ldots & d_{2n} \\ \ldots & \ldots & \ldots \\ d_{n1} & d_{n2} \ldots & d_{nn} \end{bmatrix} \tag{A1}$$

Step 2: To normalize matrix $D_{ij}$, multiply by a factor $v$:

$$N = D_{ij} \times v \tag{A2}$$

where $v = \frac{1}{\max\limits_{1 \leq i \leq n} .} \left( \sum_{j=i}^{n} d_{ij} \right), (i, j = 1, 2, \ldots, n)$.

Step 3: Develop total relation matrix $T_R$ using the normalized matrix $N$ and the identity matrix $I$:

$$T_R = N(I - N)^{-1} \tag{A3}$$

Step 4: Calculate the summation of rows ($S_R$) and columns ($S_c$) to obtain the centrality $C_E$ and causality $C_A$ from matrix $T_R$ per Equations (A4) and (A5). Then, create a causal diagram by mapping $C_E$ and $C_A$.

$$\left\{ \begin{aligned} S_R &= \left[ \sum_{j=1}^{n} T_r \right]_{n \times 1}, \quad (i = 1, 2, \ldots, n) \\ S_C &= \left[ \sum_{j=1}^{n} T_r \right]_{1 \times n}, \quad (j = 1, 2, \ldots, n) \end{aligned} \right\} \tag{A4}$$

$$\left\{ \begin{aligned} C_E &= S_R + S_c \\ C_A &= S_R - S_c \end{aligned} \right\} \tag{A5}$$

Step 5: Define a threshold value $\lambda$ to create a relationship diagram. Calculate $\lambda$ by taking an average of all $T_R$ matrix coefficients, as portrayed in Equation (A6). Transform matrix $T_R$ into the reachability matrix $R$ by changing every element for 0 and 1, making a comparison with the threshold value. If the value is less than $\lambda$, change it to 0, and if it is greater than $\lambda$, change it to 1, as displayed in Formula (A7)

$$\lambda = \frac{\sum_{i=1}^{n} \sum_{j=1}^{n} [t_R]}{n_{t_R}} \tag{A6}$$

$$R = \left\{ \begin{aligned} 1, &\ if\ t_r \geq \lambda \\ 0, &\ if\ t_r < \lambda \end{aligned} \right\} (i, j = 1, 2, \ldots, n) \tag{A7}$$

where $t_r$ is the element of matrix $T_R$ and $n_{t_r}$ is the number of elements in the $T_R$ matrix.

*Appendix B.2. Sample Calculation of Rough Number*

Taking as an example the element $d_{23}$ from reliability matrix, considering that 5 experts' judgments are as follows, 2,2,4,3,3, i.e., $d_{23} = \{2,2,4,3,3\}$. Thus, according to Equations (10)–(16) lower and upper limits for every element are evaluated as follows:

$$Lim\ (1) = 0 \qquad \overline{Lim}(1) = \frac{2+2+4+3+3}{5} = 2.8$$

$$Lim\ (2) = \frac{2+2}{2} = 2 \qquad \overline{Lim}\ (2) = \frac{2+2+4+3+3}{5} = 2.8$$

$$Lim\,(3) = \frac{2+2+3+3}{4} = 2.5 \qquad \overline{Lim}(3) = \frac{4+3+3}{3} = 3.333$$

$$Lim\,(4) = \frac{2+2+4+3+3}{5} = 2.8 \qquad \overline{Lim}(4) = \frac{4}{1} = 4$$

$$Lim\,(5) = \frac{2+2+4+3+3}{5} = 2.8 \qquad \overline{Lim}(5) = 0$$

Then it is used Equation (11) to calculate the interval of rough number for the $d_{23}$ element,

$$\overline{RN\left(\tilde{d}_{23}\right)} = \left[d_{23}^L, d_{23}^U\right]$$

$$d_{23}^L = \frac{2+2+2.8+2.5+2.5}{5} = 2.36$$

$$d_{23}^U = \frac{2.8+2.8+4+3.33+3.33}{5} = 3.252$$

Rough number for $\tilde{d}_{23}$ element is [2.36, 3.252].

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
