# Peer review of "Evaluation of Interaction between Bridge Infrastructure Resilience Factors against Seismic Hazard"

_sustainability, doi:10.3390/su141610277_

Round 1
Reviewer 1 Report
This paper used crisp DEMATEL and rough DEMATEL methods to illustrate the main resilience factors of bridges against seismic hazards. The topic is interesting, but the main content is highly repeated with the following two references, including the abstract, introduction, and research methods.
[1] Á. F. Galaviz Román, M. S. Arif Khan and G. Kabir, "Evaluation of Interaction between Bridge Infrastructure Resilience Factors Against Seismic Hazard Hazard," 2021 Third International Sustainability and Resilience Conference: Climate Change, 2021, pp. 484-488, DOI: 10.1109/IEEECONF53624.2021.9668179.
[2] Sen, Mrinal Kanti, Subhrajit Dutta, and Golam Kabir. "Evaluation of interaction between housing infrastructure resilience factors against flood hazard based on rough DEMATEL approach." International journal of disaster resilience in the built environment (2021).
Reviewer 2 Report
The article addresses an important topic of evaluation of interaction between bridge infrastructure resilience factors against seismic hazard, which is appreciated. This paper presents a methodology to evaluate the interaction between resilience factors of bridge infrastructure against seismic hazards. In the beginning, an in-depth literature review is performed to identify the critical reliability and recovery factors that can represent the resiliency of the bridge infrastructure. The crisp DEMATEL and rough DEMATEL methods have been used to evaluate the interaction between resilience factors. Then, the causal and relationship diagrams were established for both reliability and recovery factors. The Reviewer appreciates the efforts done in this paper, however, the Reviewer has some concerns regarding to the abstract, introduction, methodology, results, discussion and references. The English language should be checked by the Native Speaker (conclusions). In Reviewer's opinion the current version of the paper should be subjected for major revised. In addition, the paper was prepared very carelessly, thus please check template of this Journal.
Introduction:
- In this part of the text please add or much more underline, what is the novelty of this research? What is the difference between this paper and other papers which were cited in the text?
- The aim of this paper should be shorter and more clearly.
- The last part of the introduction is unnecessary (line 95 - 99).
- Please describes the problem of seismic response based on the real object (bridge, structure etc)
https://doi.org/10.3390/ma14164493
https://doi.org/10.12989/scs.2022.42.6.747
https://doi.org/10.1061/(ASCE)GM.1943-5622.0002378
https://doi.org/10.1016/j.soildyn.2022.107384
https://doi.org/10.1002/stc.208
https://doi.org/10.1016/j.soildyn.2007.07.001
Methods:
· Please explain the accuracy of your method based on the other researches (papers with similar researches).
· Why do you use only these parameters from table 1 and why only 5 experts were taking into account?
Results:
· Figure 1 and 3 are not clear. Please try to show better your results (presentations of results).
· Please explain the mechanisms, tendency which had impact of the results.
Conclusion:
· Please use bullets and underline the most important conclusions from your research.
Finally, I hope that my comments will be helpful for the authors.
Reviewer 3 Report
Manuscript presented interesting results and my comments to improve the manuscript content are following:
1. In manuscript formatting and References, authors should follow the journal style.
2. Abstract not clear and authors should modify this section.
3. Please highlight the study novelty and aims in introduction section.
4. Manuscript well organized.
5. Please, any Eq. not created by authors should provide the reference.
6. The Figures quality not good and not clear, please modify all.
7. Please provide more explain for obtained results.
8. Conclusion section not well presented, not clear and very poor. Please re-write this section.
Round 2
Reviewer 1 Report
The authors have revised the discussion section of the manuscript to reduce the similarity from their former work. The novelty of this paper is not about the theory or the calculation procedure but a new application scenario. Therefore, the review should change the statement of “this paper proposes a method for evaluating …” in the paper to “this paper implements a method…”to be more precise. Other than that, this paper is valuable for publication.
Author Response
Thank you for your suggestion. We revised the manuscript accordingly.
Reviewer 2 Report
Thank you for your improving.
Author Response
Thank you for your positive feedback.
Reviewer 3 Report
Authors modified the manuscript content.
Author Response
Thank you for your positive feedback.